# Empirical evidence on factors influencing farmers' administrative burden: A structural equation modeling approach

**Christian Ritzel** [ORCID] *, **Gabriele Mack, Marco Portmann** [ORCID]**, Katja Heitkämper, Nadja El Benni**

Federal Department of Economic Affairs, Education and Research–Agroscope, Ettenhausen, Switzerland

* christian.ritzel@agroscope.admin.ch

**Data Availability Statement:** All relevant data are within the manuscript and its Supporting Information files.

## Abstract

Direct payments represent a large share of Swiss farmers' total household income but compliance with related requirements often entails a high administrative burden. This causes individuals to experience policy implementation as onerous. Based on a framework for administrative burden in citizen-state interactions, we test whether farmers' individual knowledge, psychological costs and compliance costs help to explain their perception of administrative burden related to direct payments. We refine this framework by testing different specifications of interrelations between psychological costs and perceived administrative burden based on findings from policy feedback theory and education research. Structural Equation Modeling (SEM) is applied to data collected from a representative sample of 808 Swiss farmers by postal questionnaire in 2019. We find that compliance costs and psychological costs contribute significantly to the perceived administrative burden. In contrast, farmers' knowledge level contributes to this perception not directly but indirectly, with higher knowledge reducing psychological costs. Our results support policy feedback theory, in that a high level of administrative burden increases psychological costs. Furthermore, well-educated and well-informed farmers show a more positive attitude toward agricultural policy and thus perceive administrative tasks as less onerous. Policy-makers should invest in the reduction of administrative requirements to reduce compliance costs.

## 1. Introduction

For Swiss farmers, area-based or animal-based direct payments granted by the government for the delivery of public goods such as biodiversity and fresh air, represent an important income contribution. The average annual farm income of a Swiss farmer amounts to CHF 67,190, whereby CHF 73'746 come from direct payments [1]. To reduce the negative environmental impacts of agricultural production, direct payments are linked to environmental cross-compliance standards [2]. In addition, in the last 20 years, a number of voluntary agri-environmental programs have been introduced in order to preserve biodiversity on farmland, landscape quality, or to promote animal-friendly housing systems. Farmers participating in agri-environmental programs have to meet program-specific requirements.

**Funding:** The author(s) received no specific funding for this work.

**Competing interests:** The authors have declared that no competing interests exist.

Cross-compliance standards and participation in voluntary agri-environmental schemes require a variety of proofs such as crop rotation plans, nutritional balances or records on grazing and free ranging periods in order to qualify for direct payments. Many farmers perceive their administrative workload associated with the application of direct payments as more onerous than their physical workload on the farm [3]. Indeed, various studies have shown that farmers perceive their administrative workload as a burden [4–6]. For instance, [4] found that mounting paperwork, higher workloads, and changes in agricultural regulations have measurably increased the stress levels of farmers in England and Wales. For Swiss farmers, [7] showed that high levels of administrative burden are more likely to lead to personal frustration and ultimately to burnout in the farming sector.

Research in the field of agricultural economics and policy has shown that administrative burden negatively affects farmers' health [7], reduces the effectiveness of agri-environmental programs [8–10], and negatively affects farmers' perception of government [6]. However, very little is known about why farmers perceive such a high administrative burden.

In recent years, researchers in the field of public administration have developed a framework that conceptualizes the administrative burden of citizen-state interactions [11, 12]. The framework considers three factors influencing citizens' perceived administrative burden: 1. learning costs, 2. psychological costs, and 3. compliance costs. Our study adopts this framework and aims to analyze empirically (a) how these factors influence farmers' perceived administrative burden, and (b) how the factors influence each other. Based on findings from policy feedback theory and education research, we analyze different possible interrelations between psychological costs and perceived administrative burden. Based on our results, we provide recommendations for government as to how the perceived administrative workload of farmers could be reduced. This knowledge is essential for political initiatives focused on simplifying farmers' administrative workload.

The added value of our study is twofold: First, to the best of our knowledge, the framework of administrative burden on citizen-state interactions has not been used previously in the context of agricultural policies, namely the administrative burden related to direct payments for farmers. Our study therefore aims to close scientific and political knowledge gaps by combining this theory with empirical evidence from the farming sector. Second, we refine the framework by testing different specifications of interrelations between psychological costs and the perceived administrative burden based on findings from policy feedback theory and education research. The results inform agricultural policy-makers as to how the perceived administrative burden related to direct payments can be reduced.

## 2. The framework of administrative burden

[13] define 'administrative burden' as "an individual's experience of policy implementation as onerous". This definition draws a clear distinction from burdens considered as administrative obstacles, such as formal rules. In addition, it points to the "costs that individuals experience in their interaction with the state" [13; pp. 45]. Thus, [11] identify three different categories of costs that might influence the administrative burden (Table 1). This conceptualization is

**Table 1. Factors influencing administrative burden [11].**

| Type of Cost | Examples |
| --- | --- |
| Learning costs | Individuals must learn about the program, whether they are eligible, the nature of the benefits, and how to access the program. |
| Psychological costs | Individuals face loss of autonomy or power, or an increase in stress. |
| Compliance costs | Individuals must complete forms and provide documentation. |

distinctive in that it considers not only rational but also cognitive and psychological aspects of administrative burden. The proposed concept includes findings from behavioral economics showing that individuals often do not make decisions by simply weighing costs against expected benefits, due to cognitive biases that generate a "disproportionate response to burden" [13; pp. 46]. The concept also includes findings from social psychology showing that individuals have a basic need for autonomy over their self and actions. For this reason, individuals often perceive their administrative workload as a loss of autonomy, which affects their perception of the administrative burden.

[11] subsume various costs arising from searches for information on public services under "learning costs". This category explains why factors such as low education, language barriers, and limited knowledge of other public programs often have a negative effect on the uptake of public policy programs. The authors suggest that learning costs be documented based on the public's lack of knowledge about the programs [11]. Psychological costs refer to a sense of loss of power or autonomy in interactions with the state, or the stresses of dealing with administrative processes. Compliance costs represent burdens of following administrative rules and requirements, such as costs of completing forms or documenting status. Consequently, compliance costs have to be considered as costs arising from complying with federal regulations.

## 3. Conceptual models and hypotheses

In this study, the framework developed by [11] is applied to the farming sector in Switzerland. This consists of relatively small family farms, where the farming family itself generally carries out the administrative work. We develop three conceptual models describing (i) potential relationships between the three factors and the farmers' perceived administrative burden, and (ii) potential relationships between the factors themselves. For this purpose, first, we present the basic conceptual model and derive empirically testable hypotheses (Model 1). Second, we present variants of the basic conceptual model by integrating findings from policy feedback theory and education research, and derive two further hypotheses (Model 2 and Model 3).

### 3.1. Hypotheses for the basic conceptual model (Model 1)

**3.1.1 Knowledge level.** We subsume factors such as farmers' education level and information level with regard to the cross-compliance and direct-payment system under the category "knowledge level". This category reflects "learning costs" as proposed by [11]. With regard to the relationship between 'knowledge level' and 'administrative burden', we formulate Hypothesis 1:

• A high knowledge level decreases the perceived administrative burden (H1).

This hypothesis is based on findings from medical research showing that specifically targeted education programs and training can reduce the burden on patients' caregivers [14, 15]. Therefore, it is highly likely that well-educated and well-informed farmers will perceive administrative tasks as less onerous.

**3.1.2 Compliance costs.** Especially for small businesses such as Swiss family farms that do not have a person particularly responsible for administrative and legal issues, compliance costs in the form of resources expended on meeting tax obligations are considered as very onerous [16]. Transposed to our context, it is highly likely that high compliance costs increase farmers' perceived administrative burden. Consequently, we test Hypothesis 2:

• High compliance costs increase the perceived administrative burden (H2).

### 3.1.3 Psychological costs

Based on farmers' responses to a questionnaire, [3] found that the direct-payment policy with its cross-compliance restrictions narrows farmers' entrepreneurial freedom. More precisely, farmers perceive a loss of autonomy because of the direct-payment policy, which in turn might negatively affect their perceived administrative burden. High psychological costs may aggravate stress associated with administrative tasks. Against this background, we test Hypothesis 3:

• High psychological costs increase the perceived administrative burden (H3).

Furthermore, we empirically test the relationships among the three factors influencing administrative burden. Regarding the relationship between knowledge level and compliance costs, studies in entrepreneurship research indicate that education has a strong positive effect on self-employment success [17, 18]. Experimental evidence suggests that a high information level results in more original and more appropriate solutions to problems [19]. Therefore, it is highly likely that well-educated and well-informed farmers are better able to manage their business, including efficient and effective handling of administrative tasks, which reduces compliance costs, so that we formulate Hypothesis 4:

• A high knowledge level decreases compliance costs (H4).

Moreover, we expect knowledge level to have a positive effect on psychological costs. Results from entrepreneurial research show that specific entrepreneurship education and information level play an important role in positively influencing the attitude toward starting a business [20]. In our context, well-educated and well-informed farmers may exhibit lower psychological costs. This relationship is reflected by Hypothesis 5:

• A high knowledge level decreases psychological costs (H5).

Finally, it might be assumed that compliance costs influence psychological costs. Empirical evidence in the field of political psychology reveals that compliance costs negatively influence the attitude toward a particular policy, so that "even the good Europeans become unenthusiastic about compliance when costs rise" [21]. This implies that high compliance costs increase psychological costs. In other words, the more time a farmer spends on completing and compiling evidence needed to qualify for direct payments, the more likely it is that he or she will exhibit a negative attitude toward the cross-compliance and direct-payment policy. Therefore, we test Hypothesis 6:

• High compliance costs increase psychological costs (H6).

### 3.2. Variants of the basic conceptual model (Model 2 and 3)

A literature review on potential interactions between the factors and administrative burden suggests further relationships between psychological costs and the perceived administrative burden than the one described in Model 1. We capture these findings by conceptualizing two variants of the basic conceptual model (Model 2 and 3).

Model 2 reflects the findings of policy feedback theory [13, 22, 23] and postulates that administrative burden influences the individual attitude toward a policy. In this context, a study by M reveals that a high level of administrative burden increases the probability that farmers will exhibit a negative attitude toward the cross-compliance and direct-payment policy. According to policy feedback theory and empirical evidence by [6], we test Hypothesis 3a:

• A high level of administrative burden increases psychological costs (H3a).

Model 3 combines H3 and H3a. We assume that psychological costs and administrative burden positively influence each other. In other words, formulated as Hypothesis 3b:

- Psychological costs and administrative burden are positively correlated (H3b).

Hypothesis H3b is supported by findings from education research indicating that experiences with and positive attitudes toward computers are positively correlated [24, 25].

Fig 1 shows the three different conceptual models for (a) the basic Model 1, (b) Model 2, and (c) Model 3.

The three conceptual models displayed in Fig 1 illustrate the direct effects between two variables, which will be tested based on the hypotheses outlined above. Additionally, we test for indirect and total effects. For Model 1, the following three indirect effects exist: First, the indirect effect of 'knowledge level' on 'administrative burden' with 'compliance costs' as mediator variable. Second, the indirect effect of 'knowledge level' on 'administrative burden' with 'psychological costs' as mediator variable. Third, the indirect effect of 'compliance costs' on 'administrative burden', likewise with 'psychological costs' as mediator variable. In Model 2, the second and third indirect effect of Model 1 cannot be tested, because 'administrative

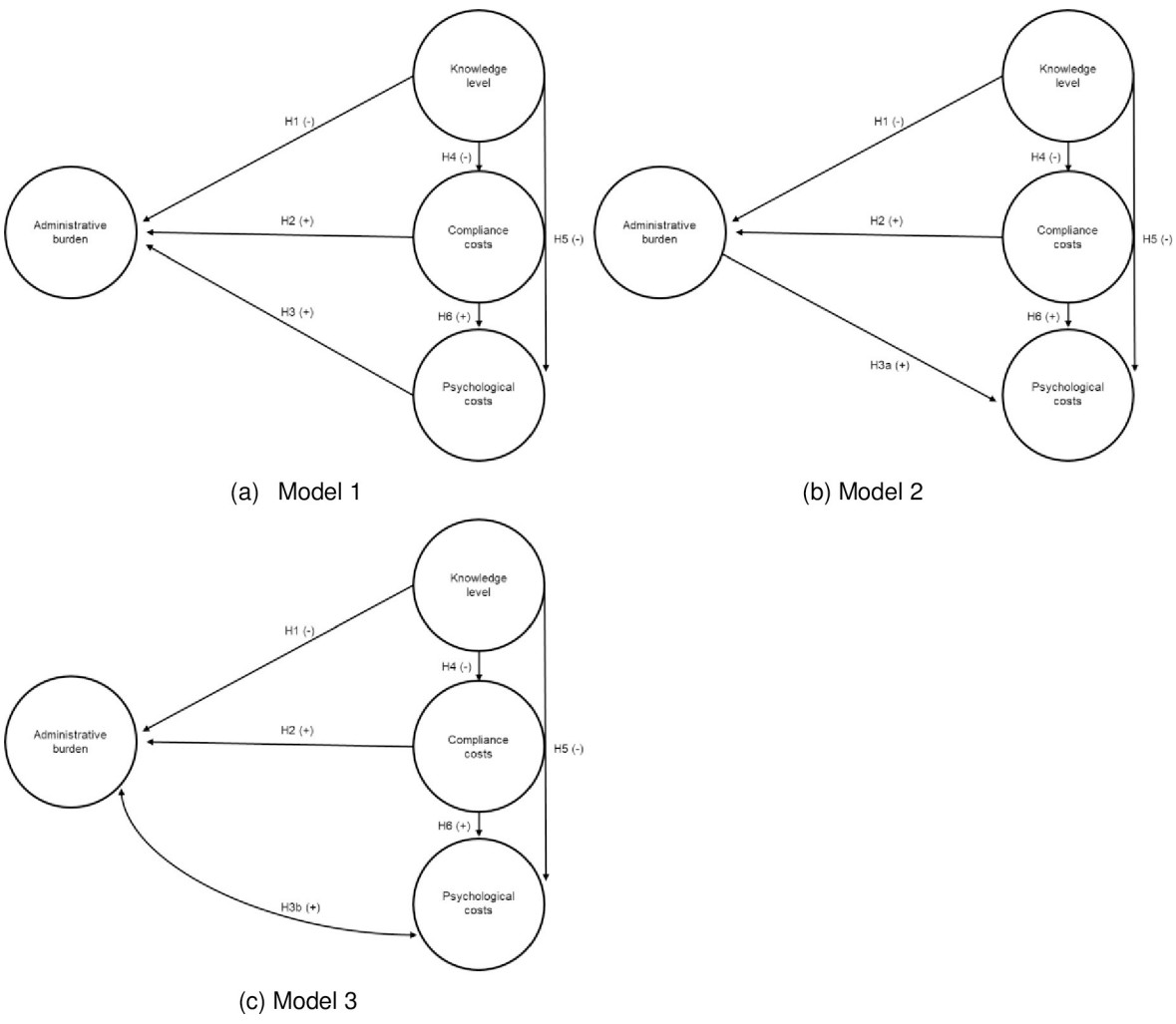

(a)  Model 1                                                        (b)  Model 2

(c)  Model 3

**Fig 1.  The three conceptual models.**

burden' serves as a predictor of 'psychological costs'. For this reason, however, the indirect effect of 'compliance costs' on 'psychological costs' with 'administrative burden' as mediator variable can be identified. For Model 3, we introduced a correlation between 'administrative burden' and 'psychological costs'. Thus, only the indirect effect of 'knowledge level' on 'administrative burden' with 'compliance costs' as mediator variable can be estimated. The total effects are calculated as the sum of direct effects plus indirect effects [26].

## 4. Materials and methods

Data from a written questionnaire are used to test the three conceptual models by structural equation modeling. The administrative burden and its three influencing factors are modeled as latent constructs, which are measured based on observed variables.

### 4.1. Database

A written survey of 2,000 randomly selected Swiss farmers was conducted from February to April 2019. Farmers' contact information was provided by the Swiss Federal Office for Agriculture, which maintains a database of all farm households that receive direct payments, comprising about 98% of Swiss farms. Farmers received a written questionnaire via postal mail. The response rate was approximately 40% (N = 808). The database is similar, in terms of region, farm type, farm size, age, and education, to the total farming population [3]. The survey contains questions on farmers' experiences with administrative requirements, farmers' individual characteristics and beliefs, and questions on their attitude toward the direct-payment and cross-compliance policy [3]. Table 2 presents a description and summary statistics of variables used for the empirical analysis.

**4.1.1 Measuring 'administrative burden'.** 'Administrative burden' is measured based on the farmers' perceived administrative burden today and compared to five years ago. We therefore asked farmers to rate two questions, each on a seven-point Likert scale (Table 2).

**4.1.2 Measuring 'compliance costs'.** 'Compliance costs' are measured based on three items: (1) farmers' self-assessments with regard to whether or not the introduction of e-government has increased compliance costs; (2) time spent on completing forms; (3) time required for direct-payment inspections. We therefore asked farmers to rate one question on the introduction of e-government on a seven-point Likert scale. Additionally, we asked farmers to rate one question on time spent providing documents on a four-point ordinal scale, and one question on time spent on direct-payment inspections on a six-point ordinal scale (Table 2).

**4.1.3 Measuring 'psychological costs'.** 'Psychological costs' are measured based on four statements related to attitude toward, identification with, and loss of freedom caused by the cross-compliance and direct-payment policy. Accordingly, we asked farmers to rate four statements, each on a seven-point Likert scale (Table 2).

**4.1.4 Measuring 'knowledge level'.** Farmers' 'knowledge level' is measured based on their education level and knowledge level regarding the cross-compliance and direct-payment policy. We therefore asked farmers one question to rate their education level on a six-point ordinal scale. Additionally, we asked farmers to rate three statements regarding their knowledge level about agricultural policy, each on a seven-point Likert scale (Table 2).

### 4.2. Structural equation modeling (SEM)

To model the psychological constructs and test the processes of the three proposed conceptual models presented in Fig 1, SEM is perfectly suited [27, 28]. Many psychological constructs such as individual administrative burden in citizen-state interactions and its factors are

**Table 2. Description and summary statistics of data used for the empirical analysis.**

| Variable | Description | Scale | Mean | Std. dev. | Obs. |
|---|---|---|---|---|---|
| **Administrative burden $\eta_1$** | | | | | |
| Administrative burden $y_1$ | How burdensome do you rate the current workload for administrative tasks? | From 1 = "not burdensome at all" to 7 = "very burdensome" | 4.9 | 1.6 | 800 |
| Administrative burden $y_2$ | How burdensome do you rate the current workload for administrative tasks compared to five years ago? | From 1 = "much less burdensome" to 7 = "much more burdensome" | 5.2 | 1.3 | 778 |
| **Compliance costs $\eta_2$** | | | | | |
| Compliance costs $y_3$ | How much has the administrative workload changed due to the switch to electronic forms? | From 1 = "much less" to 7 = "much more" | 4.2 | 1.5 | 786 |
| Compliance costs $y_4$ | How much time do you usually need to provide all documents for the direct-payment inspections? | From 1 = "less than 2 hours per inspection" to 4 = "more than 6 hours per inspection" | 2.0 | 0.9 | 794 |
| Compliance costs $y_5$ | How much time do you spend on your farm when the direct-payment inspection takes place? | From 1 = "less than 30 minutes" to 6 "more than 2.5 hours" | 4.1 | 1.2 | 795 |
| **Psychological costs $\eta_3$** | | | | | |
| Psychological costs $y_6$ | I do not identify with the federal direct-payment system. | From 1 = "not correct at all" to 7 = "fully correct" | 4.4 | 1.6 | 792 |
| Psychological costs $y_7$ | I believe that the current monitoring and inspection measures of the direct-payment system are not important. | From 1 = "not correct at all" to 7 = "fully correct" | 3.8 | 1.6 | 793 |
| Psychological costs $y_8$ | I consider the current obligations to provide proof of eligibility for direct payments as not appropriate. | From 1 = "not correct at all" to 7 = "fully correct" | 4.3 | 1.7 | 652 |
| Psychological costs $y_9$ | I feel restricted in my entrepreneurial freedom by the current direct-payment monitoring and inspection system. | From 1 = "not correct at all" to 7 = "fully correct" | 4.5 | 1.9 | 797 |
| **Knowledge level $\xi_1$** | | | | | |
| Level of education $x_1$ | • No vocational education and training<br>• Vocational education and training (VET): federal VET certificate<br>• Vocational education and training (VET): federal VET diploma<br>• Federal diploma of professional education and training (PET)<br>• Advanced federal diploma of professional education and training<br>• Bachelor, Master or higher degree of the farm manager | From 1 = "No vocational education and training" to 6 = "Bachelor, Master or higher degree of the farm manager" | 3.6 | 1.2 | 784 |
| Level of information $x_2$ | I am well-informed on current direct-payment control measures. | From 1 = "not correct at all" to 7 = "fully correct" | 4.6 | 1.4 | 797 |
| Level of information $x_3$ | I am well-informed on current obligations recording farm data. | From 1 = "not correct at all" to 7 = "fully correct" | 4.8 | 1.3 | 797 |
| Level of information $x_4$ | I am well-informed on the current agricultural policy. | From 1 = "not correct at all" to 7 = "fully correct" | 4.6 | 1.3 | 789 |

unobserved or latent [29]. By applying SEM, latent variables can be measured based on variance and covariance of observed variables, and can be further brought into relation with each other [30]. Consequently, SEM elegantly bridges the gap between theory and empirics [31]. Testing complex hypotheses makes SEM attractive for a wide range of academic fields. These include psychological research in a narrow sense [32], disciplines with a focus on human psychology, such as marketing and consumer research [33, 34], and disciplines with an organizational perspective such as management research [35, 36]. Moreover, SEM is widely applied in natural sciences such as ecological and evolutionary biology [37]. In this context, [38] highlight that SEM provides a powerful framework for promoting interdisciplinary research and holistic and integrative thinking.

In principle, SEM relies on the following two model factors [39]: First, a measurement model that measures the latent variables based on variance and covariance of observed variables. Second, a causal structural model that estimates linear relationships between different latent variables based on regression. According to [40], in SEM, the term 'causal modeling' is somewhat misleading. Rather, 'causal modeling' captures the intent of the research methodology, which is to hypothesize and specify the interrelatedness of latent variables. Thus, SEM is a confirmatory method aimed at testing proposed theories.

In this context, a distinction is made between endogenous and exogenous latent variables. Endogenous latent variables are considered as dependent variables, which are explained by at least one other (endogenous or exogenous) latent variable. In contrast, exogenous latent variables are strictly considered as explanatory variables [41]. In our case, 'administrative burden', 'compliance costs', and 'psychological costs' represent endogenous latent variables, while 'knowledge level' represents an exogenous latent variable.

Structural equation models are usually illustrated as path-diagrams (a presentation of the three applied path diagrams can be found in S1 Fig; note that, for simplification, error terms are not depicted). Latent endogenous and exogenous variables are depicted in circles and observed variables in boxes. Single-headed arrows indicate (i) the estimated impacts of coefficients obtained from the measurement model, and (ii) the estimated impacts of coefficients from the causal structural model. In the case of the measurement model, estimated parameters are considered as loadings. Two-headed curved arrows show estimated covariance, commonly interpreted as correlation [42].

According to [43], the equations of the measurement models for the endogenous and the exogenous latent variables can be formalized by Eqs (1) and (2), respectively:

$$\mathbf{y} = \Lambda_y \boldsymbol{\eta} + \boldsymbol{\varepsilon} \tag{1}$$

$$\mathbf{x} = \Lambda_x \boldsymbol{\xi} + \boldsymbol{\delta} \tag{2}$$

Where $\mathbf{y}$ is a vector of $\mathbf{p} \times \mathbf{1}$ observed variables and $\mathbf{x}$ is a vector of $\mathbf{q} \times \mathbf{1}$ observed variables. $\Lambda_y$ is the $\mathbf{p} \times \mathbf{m}$ matrix of coefficients (or loadings) $\lambda_y$ of $\mathbf{y}$ on $\boldsymbol{\eta}$ and $\Lambda_x$ is the $\mathbf{q} \times \mathbf{m}$ matrix of coefficients (or loadings) $\lambda_x$ of $\mathbf{x}$ on $\boldsymbol{\xi}$. $\boldsymbol{\eta}$ is a $\mathbf{m} \times \mathbf{1}$ random vector of endogenous latent variables and $\boldsymbol{\xi}$ is a $\mathbf{n} \times \mathbf{1}$ random vector of exogenous latent variables. $\boldsymbol{\delta}$ and $\boldsymbol{\varepsilon}$ are $\mathbf{q} \times \mathbf{1}$ and $\mathbf{p} \times \mathbf{1}$ vectors of measurement errors in $\mathbf{x}$ and $\mathbf{y}$, respectively. For a detailed description of latent and observed variables, see Table 2.

The (causal) structural model that estimates the relationship between latent variables can be formalized as follows:

$$\boldsymbol{\eta} = \mathbf{B}\boldsymbol{\eta} + \Gamma\boldsymbol{\xi} + \zeta \tag{3}$$

Where $\mathbf{B}$ is the $\mathbf{m} \times \mathbf{m}$ matrix of regression coefficients $\boldsymbol{\beta}$ related to endogenous latent variables and $\Gamma$ is the $\mathbf{m} \times \mathbf{n}$ matrix related to the coefficients $\gamma$ of the exogenous latent variables. $\zeta$ depicts a $\mathbf{m} \times \mathbf{m}$ vector of error terms.

Eqs (1) to (3) represent the general framework of a structural equation model. We test the hypothesis based on the three model variants. All of the variants capture relationships among latent variables shown in Fig 1; however, hypotheses H3 (Model 1), H3a (Model 2) and H3b (Model 3) are tested separately. Model variants are compared by using comparative model fit criteria such as the Akaike Information Criterion (AIC), Bayesian Information Criterion (BIC), and likelihood-ratio test. Models with lower values with regard to AIC, BIC, and likelihood-ratio test perform better than models with higher values [44].

By way of example, the specification for Model 1 in terms of Eqs (1) to (3) can be written as follows:

$$
\begin{bmatrix} y_1 \\ y_2 \\ y_3 \\ y_4 \\ y_6 \\ y_7 \\ y_8 \\ y_9 \end{bmatrix} = \begin{bmatrix} 1 & 0 & 0 \\ \lambda_{y2,1} & 0 & 0 \\ 0 & 1 & 0 \\ 0 & \lambda_{y4,2} & 0 \\ 0 & \lambda_{y5,2} & 0 \\ 0 & 0 & 1 \\ 0 & 0 & \lambda_{y7,3} \\ 0 & 0 & \lambda_{y8,3} \\ 0 & 0 & \lambda_{y9,3} \end{bmatrix} \begin{bmatrix} \eta_1 \\ \eta_2 \\ \eta_3 \end{bmatrix} + \begin{bmatrix} \varepsilon_1 \\ \varepsilon_2 \\ \varepsilon_3 \\ \varepsilon_4 \\ \varepsilon_6 \\ \varepsilon_7 \\ \varepsilon_8 \\ \varepsilon_9 \end{bmatrix} \tag{4}
$$

$$
\begin{bmatrix} x_1 \\ x_2 \\ x_3 \\ x_4 \end{bmatrix} = \begin{bmatrix} 1 \\ \lambda_{x2,1} \\ \lambda_{x3,1} \\ \lambda_{x4,1} \end{bmatrix} [\xi_1] + \begin{bmatrix} \delta_1 \\ \delta_2 \\ \delta_3 \\ \delta_4 \end{bmatrix} \tag{5}
$$

$$
\begin{bmatrix} \eta_1 \\ \eta_2 \\ \eta_3 \end{bmatrix} = \begin{bmatrix} \beta_{1,1} & \beta_{1,2} & 0 \\ 0 & 0 & 0 \\ 0 & 0 & \beta_{3,3} \end{bmatrix} \begin{bmatrix} \eta_1 \\ \eta_2 \\ \eta_3 \end{bmatrix} + \begin{bmatrix} \gamma_{1,1} & 0 & 0 \\ 0 & \gamma_{2,2} & 0 \\ 0 & 0 & \gamma_{3,3} \end{bmatrix} [\xi_1] + \begin{bmatrix} \zeta_1 \\ \zeta_2 \\ \zeta_3 \end{bmatrix} \tag{6}
$$

Where all variables are as previously defined. Eqs (4) and (5) refer to the measurement models for the endogenous and exogenous latent variables, and Eq (6) refers to the (causal) structural model. Our empirical structural equation model relies solely on ordinal-scaled observed variables (Table 2). Therefore, to estimate the conceptual models, we use the gsem (Generalized Structural Equation Modeling) command implemented in Stata 16 [45]. In Stata's gsem, observed items are continuous, binary, ordinal, count, or multinomial. In contrast, in sem, observed items are continuous. Models comprise linear regression, gamma regression, logit, probit, ordinal logit, ordinal probit, Poisson, negative binomial, multinomial logit, and more. gsem does not provide overall model fit criteria such as Comparative Fit Index (CFI), root mean squared error of approximation (RMSEA), or standardized root mean squared residuals (SRMR). If gsem is applied, it is unfeasible to report standardized coefficients. For a detailed description of similarities and dissimilarities between sem and gsem, see [45]. Estimated coefficients of Eqs (4) to (6) are based on Maximum-Likelihood. Standard errors of the coefficients are computed based on the Observed Information Matrix [46]. As an optimization technique for the Maximum-Likelihood estimations, we choose the Berndt–Hall–Hall–Hausman maximization algorithm [47]. To obtain indirect and total effects of the (causal) structural model, we compute non-linear combinations of coefficients [48]. As a robustness check, we estimated our three different conceptual models with sem. Corresponding results for the

(causal) structural model can be found in S1 Table (direct effects) and S2 Table (indirect and total effects). Results for the measurement models can be found in S3 Table.

## 5. Results and discussion

### 5.1. Direct effects

Table 3 presents the direct effects of the (causal) structural model. The results of the measurement models can be found in S4 Table. The output of gsem reports unstandardized coefficients, which show (i) the direction of an effect (positive or negative), and (ii) the effect strength. With regard to comparative model fit criteria AIC, BIC, and likelihood-ratio test, all three conceptual models perform equally.

'Knowledge level' shows the expected negative effect on 'administrative burden'. However, for Model 1, the effect is not statistically significant. In contrast, for Model 2 and Model 3, the negative effect is statistically significant. Therefore, H1 cannot be rejected.

For all model variants, we find the expected statistically significant positive effect of 'compliance costs' on 'administrative burden'. This implies that farmers who spend more time on providing evidence and on direct-payment inspections taking place at their farm exhibit a higher administrative burden. Consequently, high compliance costs cause administrative burden and H2 cannot be rejected.

**Table 3.  Direct effects of the (causal) structural model (unstandardized coefficients).**

| Path | Model 1 | Model 2 | Model 3 |
|---|---|---|---|
|  | GSEM | GSEM | GSEM |
| Knowledge level → administrative burden (H1) | -0.122 | -0.887* | -0.888* |
|  | (0.472) | (0.517) | (0.516) |
| Compliance costs → administrative burden (H2) | 1.305*** | 1.451*** | 1.454*** |
|  | (0.260) | (0.265) | (0.266) |
| Psychological costs → administrative burden (H3) | 0.263*** |  |  |
|  | (0.100) |  |  |
| Administrative burden → psychological costs (H3a) |  | 0.172*** |  |
|  |  | (0.064) |  |
| Administrative burden ↔ psychological costs (H3b) |  |  | 0.691** |
|  |  |  | (0.280) |
| Knowledge level → compliance costs (H4) | -0.046 | -0.048 | -0.046 |
|  | (0.230) | (0.230) | (0.230) |
| Knowledge level → psychological costs (H5) | -2.912*** | -2.769*** | -2.912*** |
|  | (0.784) | (0.737) | (0.754) |
| Compliance costs → psychological costs (H6) | 0.566*** | 0.315** | 0.566*** |
|  | (0.107) | (0.139) | (0.107) |
| **Comparative model fit criteria** |  |  |  |
| AIC | 31,663 | 31,663 | 31,663 |
| BIC | 32,089 | 32,089 | 32,089 |
| Likelihood-ratio test | -15,740 | -15,740 | -15,740 |

* $p \leq 0.1$

** $p \leq 0.05$

*** $p \leq 0.01$.

Standard errors based on Observed Information Matrix (OIM) in parentheses.

[11] hypothesize a positive effect of 'psychological costs' on 'administrative burden' (H3). This implies that farmers with a negative attitude toward the cross-compliance and direct-payment policy perceive administrative tasks as more onerous. The positive effect of 'psychological costs' on 'administrative burden' is statistically significant. Therefore, H3 cannot be rejected. In contrast to [11], Policy Feedback Theory hypothesizes a positive effect of 'administrative burden' on 'psychological costs'. Likewise, the underlying hypothesis H3a cannot be rejected. We find a statistically significant positive effect of 'administrative burden' on 'psychological costs'. Based on H3 and H3a, we formulated H3b. Statistically significant findings indicate that 'administrative burden' and 'psychological costs' are positively correlated. In other words, administrative burden and psychological costs reinforce each other. Consequently, H3b cannot be rejected.

As expected, the effect of the exogenous latent variable 'knowledge level' on 'compliance costs' is negative. However, for all model variants, the effect is not statistically significant. Thus, H4 has to be rejected. In contrast, H5 cannot be rejected. As hypothesized, the effect of 'knowledge level' on 'psychological costs' is statistically significantly negative for all model variants. This implies that farmers with a high knowledge level tend to have lower psychological costs. In other words, farmers who are well-educated and well-informed tend to have a significantly more positive attitude toward and a stronger identification with the cross-compliance and direct-payment policy. Furthermore, policy-supporting farmers with a high knowledge level do not feel restricted in their entrepreneurial freedom.

Finally, as expected, 'compliance costs' increase 'psychological costs'. More specifically, farmers who spend more time on administrative requirements tend to show a more negative attitude toward the cross-compliance and direct-payment system. Likewise, the perception that the switch to e-government has increased administrative workload causes high psychological costs. For all model variants, the positive effect of 'compliance costs' on 'psychological costs' is statistically significant. Therefore, H6 cannot be rejected.

## 5.2. Indirect and total effects

Table 4 reports indirect and total effects of the (causal) structural model. Results of total effects are reported using the same pattern as for results of indirect effects.

For all model variants, 'knowledge level' negatively influences 'administrative burden' through the mediator variable 'compliance costs'. However, in none of the model variants is this effect statistically significant. Even though psychological costs increase farmers' administrative burden, the indirect effect of 'knowledge level' on 'administrative burden' with 'psychological costs' as mediator variable is statistically significantly negative. This implies that, first, a high knowledge level reduces psychological costs. Consequently, a positive attitude toward and a strong identification with the cross-compliance and direct-payment policy leads to farmers perceiving administrative tasks as less onerous. The indirect effect of 'compliance costs' on 'administrative burden' with 'psychological costs' as mediator variable is statistically significantly positive. Findings suggest that, first, high compliance costs cause psychological costs to increase; ultimately, high psychological costs lead in turn to administrative burden. The indirect effect of 'compliance costs' on 'psychological costs' with 'administrative burden' as mediator variable is statistically significantly positive (Model 2). This indicates that, first, high compliance costs cause administrative burden. Ultimately, this leads to a negative attitude toward and a lack of identification with the cross-compliance and direct-payment policy.

The total effect of 'knowledge level' on 'administrative burden' (with 'compliance costs' as mediator variable) represents a significant negative effect for Model 2 and Model 3. Therefore, farmers with a high knowledge level indicate lower levels of administrative burden. For Model

**Table 4. Indirect and total effects of the (causal) structural model (unstandardized coefficients).**

| Indirect effects | Model 1 | Model 2 | Model 3 |
|---|---|---|---|
| | GSEM | GSEM | GSEM |
| Knowledge level → administrative burden | -0.060 | -0.069 | -0.067 |
| Mediator variable: compliance costs | (0.300) | (0.335) | (0.334) |
| Knowledge level → administrative burden | -0.766** | | |
| Mediator variable: psychological costs | (0.355) | | |
| Compliance costs → administrative burden | 0.149*** | | |
| Mediator variable: psychological costs | (0.052) | | |
| Compliance costs → psychological costs | | 0.249*** | |
| Mediator variable: administrative burden | | (0.087) | |
| **Total effects** | **Model 1** | **Model 2** | **Model 3** |
| | GSEM | GSEM | GSEM |
| Knowledge level → administrative burden | -0.183 | -0.956* | -0.955* |
| Mediator variable: compliance costs | (0.475) | (0.508) | (0.507) |
| Knowledge level → administrative burden | -0.888* (0.516) | | |
| Mediator variable: psychological costs | | | |
| Compliance costs → administrative burden | 1.455*** | | |
| Mediator variable: psychological costs | (0.266) | | |
| Compliance costs → psychological costs | | 0.564*** | |
| Mediator variable: administrative burden | | (0.107) | |

\* $p \leq 0.1$

\*\* $p \leq 0.05$

\*\*\* $p \leq 0.01$.

Standard errors based on delta method in parentheses.

1, the total effect of 'knowledge level' on 'administrative burden' (with 'psychological costs' as mediator variable) is likewise significantly negative. Here too, farmers with a high knowledge level perceive administrative tasks as less onerous. Likewise, for Model 1, the total effect of 'compliance costs' on 'administrative burden' is statistically significantly positive. Compliance costs in the form of time spent gathering direct-payment evidence and the perceived increase in administrative workload due to the switch to e-government intensify administrative burden. In the case of Model 2, the total effect of 'compliance costs' on 'psychological costs' is statistically significantly positive. Consequently, high compliance costs cause psychological costs to increase.

## 6. Conclusions and policy recommendations

The administrative burden in citizen-state interactions can be tackled in the realm of agricultural policy! By applying SEM, we are able to refine and test the theoretical framework developed by [11] in the context of the Swiss cross-compliance and direct-payment policy. We model farmers' administrative burden and factors affecting it as latent constructs based on observed variables. In general, we find that knowledge level, compliance and psychological factors explain farmers' administrative burden. The results of all three models confirm that not only rational factors such as compliance costs but also psychological factors influence farmers' perceived administrative burden. Additionally, we find a mutual positive relationship between psychological factors and perceived administrative burden, which highlights the importance of political attitude for farmers' perceived administrative burden. This result also suggests that

policy feedback theory is a valuable extension of the framework of administrative burden. Interestingly, farmers' knowledge level tends to affect the perceived administrative burden not directly but indirectly, as a high knowledge level reduces psychological costs

Direct payments represent a large share of Swiss farmers' total household income. Furthermore, voluntary agri-environmental direct-payment schemes in particular require compliance with additional standards, and administrative burden may hinder widespread adoption. Therefore, to increase acceptance of the cross-compliance and direct-payment policy, reducing farmers' administrative burden is of crucial importance for agricultural policy-makers. The present study stresses the importance of education and information in reducing psychological costs. Well-educated and well-informed farmers exhibit lower psychological costs and perceive administrative tasks as less onerous. In other words, the level of information on the cross-compliance and direct-payment policy is the primary factor that positively influences farmers' attitudes toward the policy. Since we find a strong positive effect of compliance costs on administrative burden, policy should focus on a successive reduction of administrative requirements. In this context, younger farmers in particular should be better able to cope with new information technologies supporting the efficient handling of administrative requirements, while older farmers should be systematically trained and supported. Thus, political initiatives to reduce farmers' administrative burden should focus on the one hand on measures that decrease compliance costs. Examples might include reducing the number of application documents to be completed for direct payments, or investing in e-government services. On the other hand, political initiatives focusing on a positive attitude toward agricultural policy could also help to decrease farmers' perceived administrative burden. Consequently, to tackle farmers' administrative burden effectively, policy measures and agricultural extension services should aim to increase investments in education and training, especially targeting the handling of administrative requirements.

## Supporting information

**S1 Data.**
(XLS)

**S1 Fig. Path diagrams of the three applied structural equation models.**
(TIF)

**S1 Table. Direct effects of the (causal) structural model (unstandardized coefficients).**
(DOCX)

**S2 Table. Indirect and total effects of the (causal) structural model (unstandardized coefficients).**
(DOCX)

**S3 Table. The measurement models (unstandardized coefficients).**
(DOCX)

**S4 Table. The measurement models (unstandardized coefficients).**
(DOCX)

## Author Contributions

**Conceptualization:** Christian Ritzel, Gabriele Mack.

**Data curation:** Gabriele Mack, Katja Heitkämper.

**Formal analysis:** Christian Ritzel, Gabriele Mack.

**Funding acquisition:** Gabriele Mack.

**Investigation:** Christian Ritzel.

**Methodology:** Christian Ritzel, Gabriele Mack, Marco Portmann, Nadja El Benni.

**Project administration:** Gabriele Mack.

**Resources:** Gabriele Mack.

**Software:** Christian Ritzel.

**Supervision:** Gabriele Mack, Nadja El Benni.

**Validation:** Christian Ritzel, Gabriele Mack, Marco Portmann, Nadja El Benni.

**Visualization:** Christian Ritzel.

**Writing – original draft:** Christian Ritzel, Gabriele Mack, Nadja El Benni.

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
