## [Decision Letter · Decision Letter 0]

19 Aug 2020

PONE-D-20-17818

Empirical evidence on factors influencing farmers’ administrative burden: A structural equation modeling approach

PLOS ONE

Dear Dr. Ritzel,

Thank you for submitting your manuscript to PLOS ONE. After careful consideration, we feel that it has merit but does not fully meet PLOS ONE’s publication criteria as it currently stands. Therefore, we invite you to submit a revised version of the manuscript that addresses the points raised during the review process.

Methodological issues raised by Reviewer 1 need to be addressed. In addition, the discussion section needs to explain the limitations of the measurement and structural components of the structural equation models and discuss possible ways to improve model fit in future studies.

We look forward to receiving your revised manuscript.

Kind regards,

Asim Zia, Ph.D.

Academic Editor

PLOS ONE

Additional Editor Comments:

Methodological issues raised by Reviewer 1 need to be addressed. In addition, the discussion section needs to explain the limitations of the measurement and structural components of the structural equation models and discuss possible ways to improve model fit in future studies.

2. For this observational, survey-based study, please avoid causal-sounding language (such as 'impact', 'effect', or 'influence') when reporting associations.

Reviewers' comments:

Reviewer's Responses to Questions

**Comments to the Author**

1. Is the manuscript technically sound, and do the data support the conclusions?

Reviewer #1: Partly

2. Has the statistical analysis been performed appropriately and rigorously? 

Reviewer #1: Yes

3. Have the authors made all data underlying the findings in their manuscript fully available?

Reviewer #1: No

4. Is the manuscript presented in an intelligible fashion and written in standard English?

Reviewer #1: Yes

5. Review Comments to the Author

Reviewer #1: This study sought to examine the relationship between psychological and compliance costs and perceived administrative burden related to direct payments for Swiss farmers. This is a well written paper in need of just minor revisions/clarifications. These suggested changes are provided below.

One minor issue is that direct payments is not exactly defined. While it may be painfully obvious to the authors what this means, it would be helpful to outside readers less familiar with Swiss farming to clarify what is meant by direct payments.

The authors state that one of the added benefits of their study is that they are able to refine the framework of administrative burden on Swiss farmers by testing three theoretical models. However, it is not made clear which model the authors determine to be the most appropriate for explaining these interrelationships.

The authors also state on page 13 that their specific hypotheses are tested separately (via models 1-3). However, the model fit criteria in Table 1 is exactly the same across all 3 models. Additionally, in the S1 Figure, the title is “Path diagram of the applied structural equation model”, these two things lead me to believe that a single model (displayed in the S1 Figure) was fit, rather than three separate models.

If three separate models were in fact examined, while it is important to interpret the results from these various models, it would be helpful if the researchers clarified their conclusion about which model fit best and was the driving force behind their recommendations to policy makers regarding the need to decrease psychological costs and thus the perceived administrative burdens of direct payment for Swiss farmers. Perhaps it would be most appropriate to include the full path diagram for this final model (including path coefficients, factor loadings, correlations, etc.) rather than the current path diagram included in the S1 Figure.

X4 is missing from the path diagram in the S1 Figure.

One final concern I have is with the specification of the measurement model. I am interested to further understand the authors’ decision to constrain one of the loading for each factor to 1 for model identification rather than estimating each of these factor loadings and imposing other constraints.

6. PLOS authors have the option to publish the peer review history of their article (what does this mean?). If published, this will include your full peer review and any attached files.

Reviewer #1: No

---

## [Author Response · Author response to Decision Letter 0]

14 Sep 2020

Response to Reviewer

Reviewer #1: This study sought to examine the relationship between psychological and compliance costs and perceived administrative burden related to direct payments for Swiss farmers. This is a well written paper in need of just minor revisions/clarifications. These suggested changes are provided below.

• One minor issue is that direct payments is not exactly defined. While it may be painfully obvious to the authors what this means, it would be helpful to outside readers less familiar with Swiss farming to clarify what is meant by direct payments.

Response: Many thanks for this comment! Right at the beginning of the introduction, we now provide a short definition on direct payments. 

• The authors state that one of the added benefits of their study is that they are able to refine the framework of administrative burden on Swiss farmers by testing three theoretical models. However, it is not made clear which model the authors determine to be the most appropriate for explaining these interrelationships.

Response: The AIC, BIC and likelihood-ratio test values do not differ across model variants. Therefore, (unfortunately) comparative fit measures do not allow determining the most appropriate model (all models “work” equally well). 

• The authors also state on page 13 that their specific hypotheses are tested separately (via models 1-3). However, the model fit criteria in Table 1 is exactly the same across all 3 models. Additionally, in the S1 Figure, the title is “Path diagram of the applied structural equation model”, these two things lead me to believe that a single model (displayed in the S1 Figure) was fit, rather than three separate models.

Response: We re-estimated the three different models to get sure that we provide the correct AIC, BIC and likelihood-ratio test values. However, the AIC, BIC and likelihood-ratio test values still do not differ across model variants. To avoid confusion, we now present the three different applied models separately in S1 Figure. 

• If three separate models were in fact examined, while it is important to interpret the results from these various models, it would be helpful if the researchers clarified their conclusion about which model fit best and was the driving force behind their recommendations to policy makers regarding the need to decrease psychological costs and thus the perceived administrative burdens of direct payment for Swiss farmers. Perhaps it would be most appropriate to include the full path diagram for this final model (including path coefficients, factor loadings, correlations, etc.) rather than the current path diagram included in the S1 Figure.

Response: As already mentioned above, unfortunately, comparative fit measures do not differ across model variants. Therefore, we are not able to conclude which model is the most appropriate one & we do not have a “final” model. 

• X4 is missing from the path diagram in the S1 Figure.

Response: Thanks for this hint! We have added x4 to the figure. 

• One final concern I have is with the specification of the measurement model. I am interested to further understand the authors’ decision to constrain one of the loading for each factor to 1 for model identification rather than estimating each of these factor loadings and imposing other constraints.

Response: Stata automatically constraints one estimator (factor loading) of the measurement model to the value of 1 (unstandardized coefficients). Unfortunately, by using gsem, it is not possible to report standardized coefficients as it is possible by using sem. By choosing standardized coefficients in sem, no estimator of the measurement model is constraint to 1. In our opinion, the impossibility of reporting standardized coefficients is one «disadvantage» of the gsem command.

---

## [Editor Report · Decision Letter 1]

8 Oct 2020

Empirical evidence on factors influencing farmers’ administrative burden: A structural equation modeling approach

PONE-D-20-17818R1

Dear Dr. Ritzel,

We’re pleased to inform you that your manuscript has been judged scientifically suitable for publication and will be formally accepted for publication once it meets all outstanding technical requirements.

Kind regards,

Asim Zia, Ph.D.

Academic Editor

PLOS ONE
---

## [Editor Report · Acceptance letter]

13 Oct 2020

PONE-D-20-17818R1 

Empirical evidence on factors influencing farmers’ administrative burden: A structural equation modeling approach 

Dear Dr. Ritzel:

I'm pleased to inform you that your manuscript has been deemed suitable for publication in PLOS ONE. Congratulations! Your manuscript is now with our production department. 

Kind regards, 

on behalf of

Professor Asim Zia 

Academic Editor

PLOS ONE